

# Identification and evaluation of the novel genes for transcript normalization during female gametophyte development in sugarcane

Maokai Yan[1,*], Xingyue Jin[2,*], Yanhui Liu[2], Huihuang Chen[2], Tao Ye[2], Zhimin Hou[2], Zhenxia Su[2], Yingzhi Chen[1], Mohammad Aslam[1], Yuan Qin[1,2] and Xiaoping Niu[1]

[1] Guangxi Key Laboratory of Sugarcane Biology, State Key Laboratory for Conservation and Utilization of Subtropical Agro-Bioresources, College of Agriculture, Guangxi University, Nanning, China
[2] Key Lab of Genetics, Breeding and Multiple Utilization of Crops, Ministry of Education, Fujian Provincial Key Laboratory of Haixia Applied Plant Systems Biology, College of Life Sciences, Fuzhou, China
* These authors contributed equally to this work.

## ABSTRACT

**Background:** Sugarcane (*Saccharum spontaneum* L.), the major sugar and biofuel feedstock crop, is cultivated mainly by vegetative propagation worldwide due to the infertility of female reproductive organs resulting in the reduction of quality and output of sugar. Deciphering the gene expression profile during ovule development will improve our understanding of the complications underlying sexual reproduction in sugarcane. Optimal reference genes are essential for elucidating the expression pattern of a given gene by quantitative real-time PCR (qRT-PCR).

**Method:** In this study, based on transcriptome data obtained from sugarcane ovule, eighteen candidate reference genes were identified, cloned, and their expression levels were evaluated across five developmental stages ovule (AC, MMC, Meiosis, Mitosis, and Mature).

**Results:** Our results indicated that *FAB2* and *MOR1* were the most stably expressed genes during sugarcane female gametophyte development. Moreover, two genes, cell cycle-related genes *REC8* and *CDK*, were selected, and their feasibility was validated. This study provides important insights into the female gametophyte development of sugarcane and reports novel reference genes for gene expression research on sugarcane sexual reproduction.

Corresponding authors
Yuan Qin, yuanqin@fafu.edu.cn
Xiaoping Niu, 407127145@qq.com

## INTRODUCTION

Sugarcane (*Saccharum spontaneum* L.) is an erect perennial plant that belongs to the Poaceae family, responsible for 80% of the world's sugar production and accounting for the fifth most valuable crop worldwide (*Zhang et al., 2018*). With high fiber content and alcohol production, sugarcane also has excellent potential in the paper industry, alternative biofuel, and animal feed. Annually, sugarcane is cultivated in more than 90 countries, with

harvested yield at 1.83 billion metric tons, and generating a revenue close to $57 billion per year. Therefore, many researchers are putting tremendous efforts into sugarcane germplasm innovation and improvement (*Yang et al., 2020*). Very little progress has been obtained through normal sexual propagation and/or hybridization due to the degeneration of sugarcane reproductive organs (*Wang et al., 2008*). Thus, unveiling the fundamental mechanism of the sugarcane reproductive developmental process by genetics, genomics, and molecular technology is becoming indispensable to develop improved varieties (*Hoang et al., 2017*). The normal development of female and male gametophyte is essential for reproduction, directly influencing crop breeding and agronomic traits (*Kawamoto et al., 2020*). Previous studies revealed that the sugarcane seed setting rate is low by sexual hybridization, and the reason behind this was that the female gametophyte of sugarcane is sterile (*Wang et al., 2020*). Therefore, understanding the underlying reasons for female gametophyte development in sugarcane is a vital endeavor. The *Saccharum spontaneum* L. genome (http://www.life.illinois.edu/ming/downloads/Spontaneum_genome/) and RNA-seq data could help us elucidate the complicated molecular mechanisms of the genetics and signal transduction networksduring female gametophyte development (*Zhang et al., 2018*). Furthermore, gene expression analysis of female gametophyte development could foster an understanding of the mechanisms underlying sugarcane reproductive development (*Wang et al., 2020*). Because of its authenticity, sensitivity and efficiency, quantitative real-time PCR (qRT-PCR) is one of the most common methods used to measure the abundance of a given gene (*Bustin & Dorudi, 2002*; *Guenin et al., 2009*). Nonetheless, the authenticity and reliability of qRT-PCR results could be easily affected by the variations such as mRNA concentration, reverse transcription efficiency, enzyme activity, among other factors (*Huggett et al., 2005*). To reduce the impacts of variations and acquire accurate results, appropriate references should be adopted as an internal control to normalize the results of the target gene. For an ideal reference gene, its transcript abundance should remain constant across all tested tissues, regardless of experimental treatments and developmental stages (*Huggett et al., 2005*). Numerous studies demonstrate that the transcription level of most reported reference genes highly depend on the experimental conditions of the tissue samples, and no such reference gene can be universally used in a wide range of tissue samples (*Guenin et al., 2009*). As a result, no reference genes can be widely applied in different experiments. Until the systematic evaluation of reference genes is completed, it cannot be used to normalize qRT-PCR in specific species and/or conditions (*Jarosova & Kundu, 2010*). Previously, several systematic research studies have been done on several commercial crops, including rice (*Narsai et al., 2010*), wheat (*Long et al., 2010*), etc. Those studies identified suitable housekeeping genes that play essential cellular roles, including 18S*rRNA*, *UBQ5* and *EF1α*, to normalize the variation, paving ways for follow-up transcriptional studies. Over the past few years, an increasing number of studies have shown that the traditional housekeeping gene could not work as stable normalization factors. Meanwhile, the selection of reference genes at the transcriptome level has found many novel genes which can be expressed stably, such as with apple (*Storch et al., 2015*), pineapple sexual organs (*Jin et al., 2020*), pear peel (*Chen et al., 2020*), and papaya (*Zhu et al., 2012*). *Jin et al. (2020)*

revealed novel identified genes *CCR* and *RPS4* behave better than the traditional housekeeping genes in pineapple ovule and stamen normalization. *Chen et al. (2020)* identified a novel gene named *NAP1* (Nucleosome Assembly Protein 1) that performed better in pear peel. Taken together, genome-wide identification work could provide new candidate reference genes and avoid the inappropriate selection of internal reference genes. However, no genome-wide or transcriptome-based reference gene selection has been conducted in *Saccharum spontaneum* L, limiting further studies on the expression normalization of many key genes (*Zhang et al., 2018*). Based on RNA-seq data, the hundreds of novel reference genes and eight traditional housekeeping genes were identified to obtain the steady expressed reference genes for normalization during sugarcane female gametophyte development. As a result of this analysis, *FAB2* and *MOR1*, which were never identified as reference genes before, were selected as optimal genes during female gametophyte development in sugarcane. Those two selected genes were further validated by normalizing the transcript abundance of two known marker genes (*CDK* and *REC8*) across all the experimental samples. To our knowledge, this study comprehensively evaluated the reference genes during the five developmental stages female gametophyte in sugarcane for the first time. This study will provide information for further normalization of key genes affecting sugarcane female gametophyte development.

## MATERIALS & METHODS

### Plant material

The sugarcane (*Saccharum spontaneum* L.) cultivar Yuetang 91-976 provided by State Key Laboratory for Conservation and Utilization of Subtropical Agro-Bioresources (Guangxi, China) was used for the experiment. When the plants reached the florescence stage, five different stages of the sugarcane female gametophyte (*i.e.*, Archesporial cell (AC), Megaspore mother cell (MMC), Meiosis, Mitosis and Mature) were collected (Fig. S3). The samples were harvested in three biological replicates and quickly frozen with liquid nitrogen before storing at −80 °C.

### RNA extraction and RNA-seq

Total RNA was isolated by the Omega Total RNA kit II (R6934–02, USA). The evaluation of RNA quality was performed by gel electrophoresis and 2,000 spectrophotometer assessment at 260 nm (NanoDrop, Thermo Fisher Scientific), and Illumina sequencing was referred to by the method of *Zhao et al. (2018)*. Each sample had three independent biological replicates, and one μg of total RNA was used for library preparation. After sequencing, the adapter sequences and low-quality reads were filtered, and the clean reads were aligned to the sugarcane reference genome by STAR v2.5.0 software. The SourceForge Subread package feature Count v1.5.0 was used for the alignment results for gene quantification.

The cDNA was synthesized using the ThermoScript RT-PCR kit (Life Technologies) in a 20 μL volume reaction under the program: 42 °C for 15 min and 85 °C for 15 s. The qRT-PCR was performed according to the SYBR Premix RT reagent kit system (TaKaRa, Dalian, China).

## Candidate gene selection and primer design

Based on the average RPKM values, the CV (*i.e.* coefficient of variation) values were calculated for each gene under the corresponding experimental stage. This resulted in 50 candidates with the low CV values (Table S1). We considered only the 18 candidates with the highest RPKM values to avoid identifying genes with absent as latent references. The name, ID and sequence information of these genes were obtained from the datas uploaded by Ming Laboratory (https://www.life.illinois.edu/ming/downloads/Spontaneum_genome/). This led to identification of 12 novel selected reference genes (*ATPase, FAB2, LOG2, MCB1, MOR1, TIV110*, VHA, *VLN3, ARM*, Expressed, *GET4, PP2A,* and *TIC110*) and six traditional genes (*EF1α, TUB6, ACT7, ACTIN, GAPDH,* and *UBC28*) (Table 1).

## qRT-PCR and statistical analysis

We used the Bio-Rad Real-time PCR (Foster, USA) for qRT-PCR with SYBR Premix Ex Taq plus (TaKaRa, China) in 20 μL volume reaction. The qRT-PCR cycle program applied in this experiment was as follows: 96 °C for 2 min, 40 cycles at 95 °C for 5 s, 60 °C for 34 s, 95 °C for 15 s. Two technical replicates and three independent biological replicates were performed for each reaction, and the negative control reactions without template were performed in parallel. According to the standard curve and formula, the correlation coefficient ($R^2$) and amplification efficiency (E) were computed: $E = [10 - (1/\text{slope}) - 1] \times 100\%$.

In order to identify whether the expression of each candidate gene was stable, three statistical software packages, geNorm (*Vandesompele et al., 2002*), NormFinder (*Andersen, Jensen & Orntoft, 2004*), and BestKeeper (*Pfaffl, 2001*), were adopted. For geNorm and NormFinder algorithms, the relative expression values of all candidate genes were calculated from the raw Ct values using the formula: $E\_\Delta Ct$, where $\Delta Ct$ = each corresponding Ct value – the minimum Ct value. Then, the resulting values were imported into NormFinder and geNorm to determine the gene expression stability. For BestKeeper analysis, the coefficient variation (CV) and standard deviation (SD) were computed using the raw Ct values. Finally, the comprehensive rankings of the optimal reference genes were acquired by calculating the geometric mean.

# RESULTS

## Candidate genes selection

Transcriptome analysis across five different developmental stages of sugarcane ovule at AC, MMC, Meiosis, Mitosis and Mature (Fig. S3) resulted in 50 candidate reference genes with the high average read per kilobase per million mapped reads (RPKM) values. SD values across all stages were calculated and then the ratio of SD and Mean Value (*i.e.*, coefficient of variation, CV) were calculated as a reference index. Since lower CV values indicate less deviation, genes with CV values less than 0.6 were picked out from 50 candidate genes. A total of eighteen genes, including twelve novel genes (*ATPase, FAB2, LOG2, MCB1, MOR1, TIV110, VHA, VLN3, ARM, Expressed, GET4, PP2A,* and *TIC110*) and six traditional genes (*EF1α, TUB6, ACT7, ACTIN, GAPDH,* and *UBC28*)

**Table 1 The primers and amplification information of candidate reference genes.**

| Gene ID/name | Genbank No. | Mean | SD | CV | Primers(F/R) | Length | E (%) | R2 |
|---|---|---|---|---|---|---|---|---|
| Sspon.03G0028120/FAB2 | MW888380 | 27.71 | 1.72 | 0.06 | ACCCTTGCTTCAAGGATCAG GTTGGACTTCCCTGCCATATAC | 103 | 1.01 | 0.998 |
| Sspon.01G0006710/LOG2 | MW888381 | 23.17 | 4.91 | 0.21 | GGAACTAGGTATGAACTGCAAGA CGGCTCTGAGAGGCAAATAA | 108 | 0.99 | 0.999 |
| Sspon.08G0004380/VHA | MW888382 | 22.25 | 4.52 | 0.20 | AATTGTCGGTGCTGTCTCTC AGCCAGCTTCTTATCCAATCC | 106 | 1.02 | 0.998 |
| Sspon.01G0037150/MCB1 | MW888383 | 21.73 | 4.16 | 0.19 | CTTGCTCTGCGGTTGTCTAT GTTTGAGCTTGAGGCATTGTT | 111 | 0.99 | 0.995 |
| Sspon.06G0026890/ATPase | MW888384 | 21.16 | 4.83 | 0.23 | CGATGCGCAAGATGAAGAATG CGTCCAGATCAAACGGTCTATT | 111 | 0.97 | 0.996 |
| Sspon.08G0021360/ARM | MW888385 | 22.44 | 5.97 | 0.27 | ACAGGGATCAGATGCACAAG GAGGGAGTCACACCGAATAAAT | 89 | 1.01 | 0.999 |
| Sspon.01G0020070/TIC110 | MW888386 | 23.79 | 3.62 | 0.15 | CAGTACCTGCTGGGCATAAG AAAGGCCAACTCCTCTTTCTC | 105 | 1.00 | 0.999 |
| Sspon.03G0026510/VLN3 | MW888387 | 26.09 | 1.89 | 0.07 | GTCGACAGGGTTGTCATAACT CCCATCTACTGCAGCATCTT | 111 | 1.04 | 0.998 |
| Sspon.03G0047080/GET4 | MW888388 | 24.52 | 7.18 | 0.29 | CGCTTCTATGCTGGTGAACT GGTTCCCTTGAGATAGGTACATT | 98 | 0.99 | 0.995 |
| Sspon.03G0004010/MOR1 | MW888389 | 25.75 | 3.11 | 0.12 | CTAGCTCAGGTGGAGAAGAATG GCAAACTTTGGAGAGGGTATTG | 101 | 0.99 | 0.998 |
| Sspon.07G0009240/EF1α | MW888390 | 24.70 | 6.09 | 0.25 | CCTCCAGGATGTGTACAAGATT ACCGAAGGTGACAACCATAC | 97 | 0.98 | 0.999 |
| Sspon.03G0016270/TUB6 | MW888391 | 21.23 | 4.59 | 0.22 | TGCATTGGTACACTGGTGAG GTACTGCTGGTACTCGGAAAC | 92 | 1.04 | 0.998 |
| Sspon.08G0001560/GAPDH | MW888392 | 19.76 | 5.71 | 0.29 | CCGTGGATGTGTCAGTTGT CCCTCAGATGCAGCCTTAATAG | 94 | 1.00 | 0.996 |
| Sspon.08G0008230/UBC28 | MW888393 | 20.74 | 5.40 | 0.26 | GGCCCTGTTGCTGAAGATATG TAGTCCGGTGGGAAGTGAATAG | 110 | 1.00 | 0.998 |
| Sspon.01G0003870/PP2A | MW888394 | 19.59 | 6.22 | 0.32 | CAGGACATATCGGAGCAGTTT GCCCAGTTATATCCCTCCATAAC | 89 | 1.01 | 0.996 |
| Sspon.01G0058300/Expressed | MW888395 | 23.37 | 7.09 | 0.30 | CAGTTGTCGAGGGCACTAAA GCTCAGCATCATTCAGTAAATCC | 96 | 1.02 | 0.998 |
| Sspon.02G0047450/ACT7 | MW888396 | 19.13 | 10.30 | 0.54 | GAGCTATGAGTTGCCTGATGT CCAGGAGCTTCCATCCTAATC | 99 | 0.99 | 0.996 |
| Sspon.01G0006610/ACT | MW888397 | 19.09 | 9.58 | 0.50 | CCAGCAGGAGTTCTACATCAG CCTCTCGATGGCTGCTTG | 100 | 1.01 | 0.996 |

**Notes:**
The name, ID and sequence information of these genes is from the Saccharum Genome Database.
(http://sugarcane.zhangjisenlab.cn/sgd/html/index.html).

were screened out (Table 1). The CV values of 12 novel genes ranged from 0.06 to 0.09, which were much lower than eight traditional reference genes (0.25–0.56) (Fig. 1A). We further calculated the relative expression level of all candidates across all five

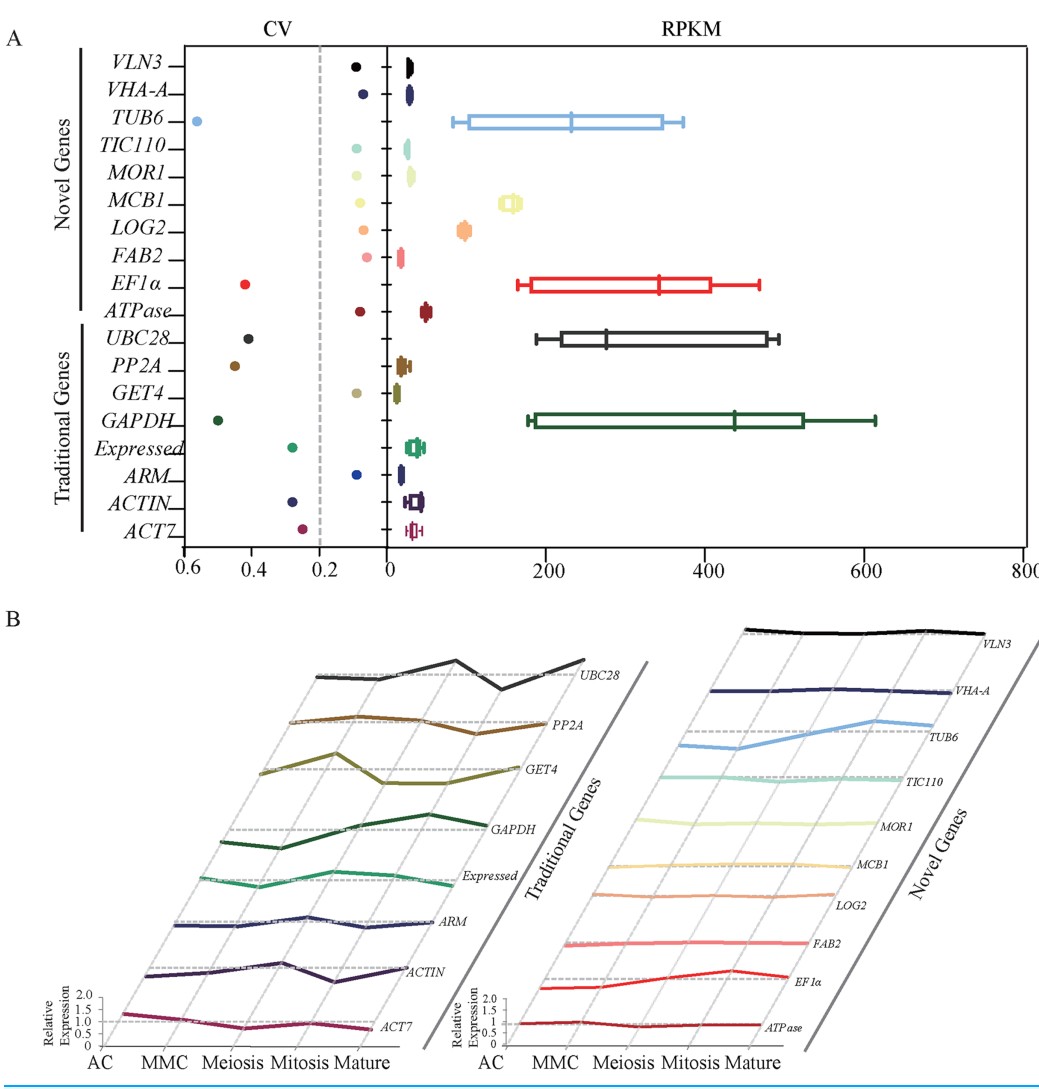

**Figure 1 Identification and evaluation of optimal reference genes based on RNA-seq data in sugarcane ovule at different developmental stages.** (A) Statistical analysis of CV and RPKM values of 18 potential candidate genes identified based on RPKM values from the RNA-seq data. The data point in scatter plot shows the variation of given gene's expression across all tested stages, the horizontal line represents CV = 0.2. Each box represents RPKM values of given gene in box and whiskers plot. (B) Relative expression level of 18 candidate genes was given by dividing RPKM values by average expression values across all sugarcane ovule developmental stages.

experimental stages to obtain more accurate data. Except for *EF1α* and *TUB6*, the novel genes showed very stable expression. In contrast, all the traditional housekeeping gene expression levels varied considerably, suggesting that the expression of novel genes may be more stable than conventional housekeeping genes during sugarcane ovule development (Fig. 1). Taken together, the results of RNA-seq showed that most of the new genes are better than traditional housekeeping genes in sugarcane ovule normalization studies.
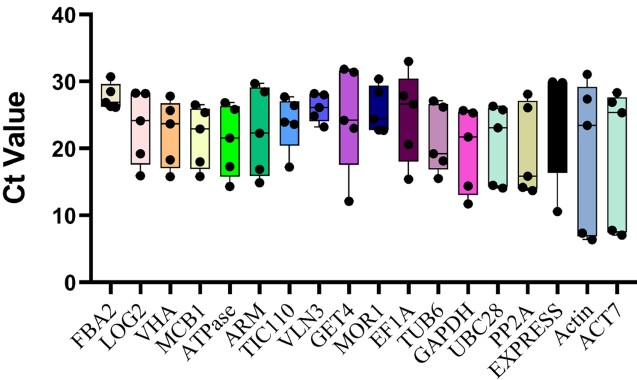

**Figure 2 Ct values of all 18 candidate genes in sugarcane ovule developmental samples.** The raw data of Ct values of 18 genes derived from all stages of sugarcane ovule developmental stages were collected and described using the box and whiskers plot. The box is determined from 25th to 75th percentiles, and the line across the box represents the median. The whiskers represent percentiles from 10th to 90th.

## PCR specificity and amplification efficiency of candidate reference genes

To investigate the transcriptional behaviors of candidate reference genes, RT-PCR and qRT-PCR experiments were adopted to confirm the primer specificity of all candidate genes. In RT-PCR results, the nonspecific amplification band and primer dimer were not observed and no signal was detected in the negative controls of all tested primers (Fig. S1). In the qRT-PCR experiment, all genes had a relatively single melting curve, indicating that the primers used in the experiment had high specificity (Fig. S2). Besides, the standard curve was established with a series of 10-fold diluted cDNA samples and then the correlation coefficient ($R^2$) and amplification efficiency (E) of each gene was calculated. The results showed that E values of all genes varied from 0.97 (*ATPase*) to 1.04 (*VLN3* and *TUB6*), and the $R^2$ of all standard curves ranged from 0.995 (*TUB6* and *GET4*) to 0.999 (*GET4, ARM, TIC110* and *EF1α*), showing their high correlation with their average value (Table 1).

## Expression profiles of candidate reference genes

The top eighteen genes were selected as candidate reference genes for qRT-PCR, and the Ct values were examined across all ovule developmental sugarcane samples to detect the transcript abundance and expression stability. The lower the Ct value means the higher the expression level, and vice versa. As showed in Fig. 2, the Ct values of the 10 novel genes ranged between 4.55 and 30.71, while the commonly used housekeeping genes, *Actin* and *ACT7*, with the broadest range 6.37 ~ 31.09 (*Actin*) and 28.36 ~ 7.05 (*ACT7*), showed the worst expression stability. These results were also consistent with their CV values, 0.54 (*Actin*) and 0.50 (*ACT7*), the two highest CV values of all candidates (Table 1). In contrast, the three novel genes *FAB2*, with the Ct values varied from 26.15 to 30.71, followed by *VLN3* (4.97 ~ 23.23) and *MOR1* (7.68 ~ 22.71), which showed minimal variation. Therefore, these genes are speculated as more suitable for normalizing sugarcane ovule development.

**Table 2** Gene expression stability ranked by geNorm, NormFinder, BestKeeper and Comprehensive ranking.

| Rank | geNorm | | NormFinder | | BestKeeper | | Comprehensive ranking | |
|---|---|---|---|---|---|---|---|---|
| | Gene | Stability | Gene | Stability | Gene | CV(%) ± SD | Gene | Geomean value |
| 1 | MCB1 | 0.15 | MOR1 | 0.09 | FBA2 | 5.48 ± 1.52 | FBA2 | 2.00 |
| 2 | LOG2 | 0.15 | FBA2 | 0.10 | VLN3 | 6.28 ± 1.63 | MOR1 | 2.00 |
| 3 | MOR1 | 0.35 | TIC110 | 0.10 | TIV110 | 11.29 ± 2.68 | MCB1 | 2.71 |
| 4 | TIC110 | 0.50 | MCB1 | 0.26 | MOR1 | 11.37 ± 2.92 | TIC110 | 3.00 |
| 5 | FBA2 | 0.60 | LOG2 | 0.41 | MCB1 | 17.72 ± 3.85 | LOG2 | 3.27 |
| 6 | VLN3 | 0.88 | VLN3 | 0.80 | VHA | 18.71 ± 4.16 | VLN3 | 3.91 |
| 7 | VHA | 1.11 | VHA | 1.12 | LOG2 | 19.28 ± 4.46 | VHA | 6.32 |
| 8 | EF1α | 1.39 | EF1α | 1.50 | ATPase | 20.32 ± 4.30 | EF1α | 8.24 |
| 9 | ATPase | 1.61 | ATPase | 1.74 | TUB6 | 20.38 ± 4.32 | ATPase | 8.32 |
| 10 | TUB6 | 1.95 | TUB6 | 2.22 | EF1α | 21.69 ± 5.35 | TUB6 | 9.32 |

## Stability evaluation by geNorm, BestKeeper and NormFinder analysis

To obtain an accurate result, three standard programs BestKeeper, geNorm and NormFinder were combined to detect the expression stability of eighteen candidate reference genes. For the geNorm algorithm, the average variation of tested genes is defined as M-values, with lower M value and more stability. As shown in Table 2, MCB1 and LOG2 showed the most stable expression level owing to their lowest M values (0.15). While the M value of TUB6 was the highest, indicating that TUB6 is the most variable among these candidates. The NormFinder algorithm is based on a mathematical model that can identify the most suitable reference gene by calculating both the overall variation of all candidates and variation between the given samples (Andersen, Jensen & Orntoft, 2004). The results demonstrated that MOR1, with the stability value of 0.09, followed by FAB2 (0.10), TIC110 (0.10), were the most stable genes. MCB1, which ranked in the first position by geNorm, was ranked in the fourth position by NormFinder. Not surprisingly, TUB6 was also evaluated as the most unreliable gene by NormFinder (Table 2). BestKeeper is also an EXCEL-based tool determining the most stable reference gene by calculating SD and CV. Both SD value and CV value were negatively correlated with gene expression stability, and thus candidate genes with lower SD value and lower CV value exhibit higher normalization reliability (Pfaffl, 2001). As shown in Table 2, FAB2 with the lowest value (CV (%) ± SD = 5.48 ± 1.52) was considered as the most reliable normalization factor, followed by VLN3 (6.28 ± 1.63), TIV110 (11.29 ± 2.68), while TUB6 (20.38 ± 4.32) followed by EF1α (21.69 ± 5.35) were identified as the most unstable normalization factors.

## Consensus ranking of candidate reference genes

Owing to different statistical approaches among the three algorithms, the best candidate genes generated by each algorithm were not for their same ranking. To obtain a comprehensive ranking, the geometric mean method was introduced to create consensus

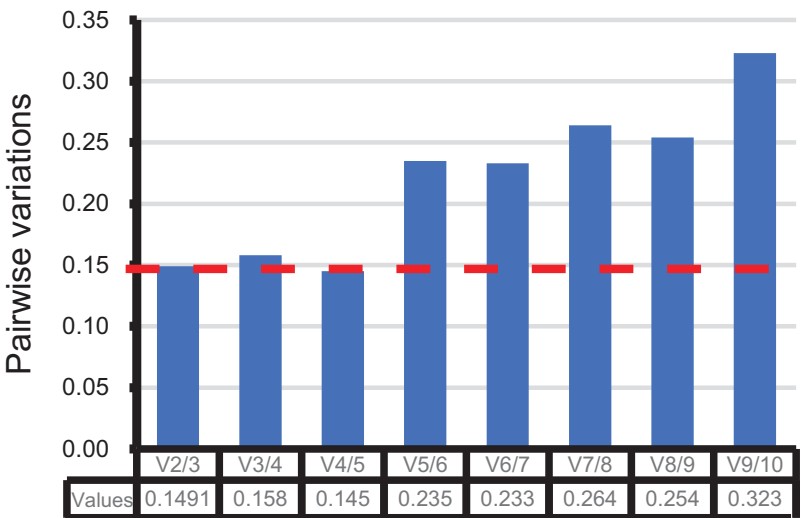

| | V2/3 | V3/4 | V4/5 | V5/6 | V6/7 | V7/8 | V8/9 | V9/10 |
|---|---|---|---|---|---|---|---|---|
| Values | 0.1491 | 0.158 | 0.145 | 0.235 | 0.233 | 0.264 | 0.254 | 0.323 |

**Figure 3 Comprehensive evaluation and selection of 10 novel genes.** Pairwise variation (Vn/Vn+1) given by geNorm. Pairwise variation values were calculated to determine the optimal number of reference genes.

results described by *Niu et al. (2015)*. As shown in Table 2, *FAB2*, *MOR1* and *MCB1* were considered as the three best genes by comprehensive analysis for sugarcane female gametophyte development.

Moreover, the pairwise variation (V) values were also calculated by geNorm to confirm the minimum number of reference genes required for accurate normalization. According to Vandesompele's research, when the V values between two adjacent normalization factors (Vn/Vn+1) were higher than the recommended value 0.15, one more reference gene than n is required (*Vandesompele et al., 2002*). As shown in Fig. 3, the first pairwise variation value is lower than the suggested value 0.15. Therefore, two top-ranked genes are enough for the reliable normalization. The third gene adopted will make no significant difference for accurate normalization across these sugarcane female gametophyte developmental stages. The results of consensus ranking showed *FAB2* and *MOR1* are the best choices for normalizing transcription during female gametophyte development in sugarcane.

## Reference gene validation

To validate the effectiveness of reference genes selected in this study, the expression of two transcripts, *CDK* and *REC8*, were analyzed by *qRT-PCR* analysis across the different ovule stages in sugarcane. *CDK* encodes a cyclin-dependent protein kinase, playing key roles in the progression of the cell proliferation or cell cycle in eukaryotes (*Endo et al., 2012*). *REC8* is reported to encode a meiotic cohesion protein that could bind sister chromatids and anchor chromosomes to the axis, thus is critical for chromosome segregation during meiosis and eukaryotes gametophyte production (*Hsieh et al., 2020*). As depicted in Fig. 4, when stable genes *FAB2* and/or *MOR1* used as the normalization factors, the expression level of *REC8* gene increased from AC stage and peaked at the MMC stage, and was reduced at the meiosis stage, followed by an increase at mitosis stage.

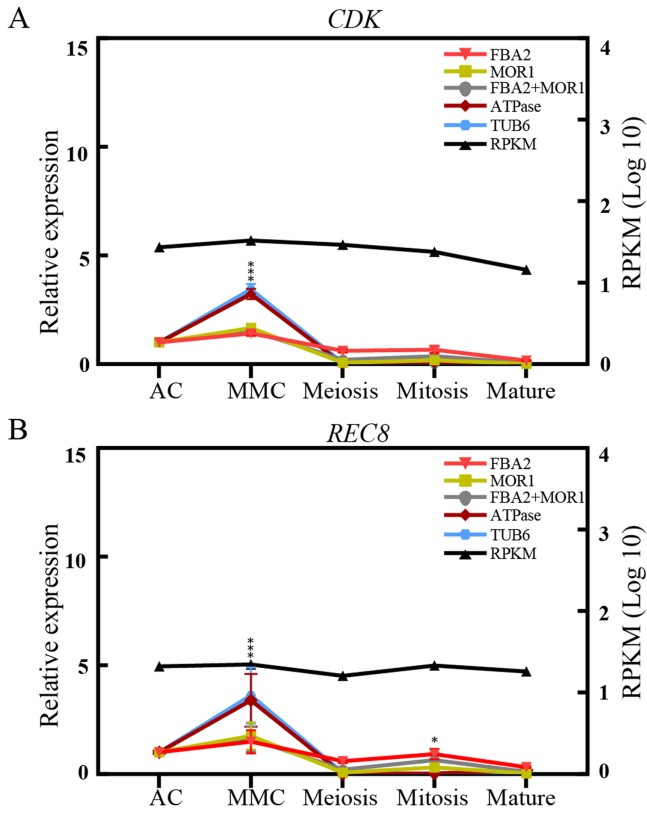

**Figure 4 Relative quantification of target genes in sugarcane ovule samples using two most stable genes and two least genes.** The relative expression levels of *CDK* (A) and *REC8* (B) normalized by the most/least stable reference genes across different sugarcane ovule development stage by qRT-PCR and RPKM values. Bars indicate the standard error (±SE) determined from three biological replicates. Three asterisks (***) indicates significant differences ($p < 0.01$), and one asterisk (*) represents differences ($p < 0.05$).

The trend of relative expression of *REC8* and *CDK* was consistent with RNA-seq data. On the contrary, when the unstable reference genes *ATPase* and/or *TUB6* were selected for validation, the expression profiles of *CDK* and *REC8* showed significant fluctuations, about 3-fold high than that of most genes at MMC stage, and the trend of expression profiles was also changed after the meiosis stage (Fig. 4).

## DISCUSSION

Due to the sterility of sugarcane female gametophyte, it is cultivated mainly by vegetative propagation in agriculture production (*Wang et al., 2008*; *Hoang et al., 2017*; *Kawamoto et al., 2020*) leading to its variety heterogeneity being reduced greatly and easily affected by abiotic and biotic factors (*Kumar et al., 2016*). To overcome said problems, efforts should be made to understand the development of the sugarcane female gametophyte. The RNA-seq and expression profiles analysis are efficient approaches to understand genes involved in the specific organ development process (*Hoang et al., 2017*). Real-time quantitative PCR (qRT-PCR) is one of the most common methods used to measure the gene expression abundance with accuracy, sensitivity, and efficiency

(*Bustin & Dorudi, 2002*; *Guenin et al., 2009*). However, the accurateness and strength of qRT-PCR results rely on the stably expressed normalization genes (*Guenin et al., 2009*; *Huggett et al., 2005*; *Jarosova & Kundu, 2010*).

   Although there have been many studies on the screening of reliable reference genes under different conditions, most of those studies focused on the traditional reference gene selections due to the deficiency of sugarcane genome and transcriptome data (*Iskandar et al., 2004*; *Ling et al., 2014*; *Guo et al., 2014*; *de Andrade et al., 2017*). For example, *eEF-1a* was used as the optimal candidate in hormone treatment experiments (*Ling et al., 2014*). *GADPH* was identified as the most suitable gene under salinity and drought conditions (*Guo et al., 2014*; *de Andrade et al., 2017*). Nonetheless, the expression of these known housekeeping genes is not stable during sugarcane ovule development (Figs. 1 and 2). RNA-seq data was introduced to select stably expressed genes under heavy metal stress and cold stress in sugarcane to broaden the candidate range of reference genes. *APRT* (*Anthranilate phosphoribosyl transferase*) was selected as the best reference gene for miRNA expression analysis under Sorghum Mosaic Virus infection and cold stress in sugarcane, respectively (*Ling et al., 2019*; *Yang et al., 2016*). In our study, reference genes were genome-widely identified based on transcriptome data across the sugarcane ovule developmental stages. A total of eighteen reference genes, including twelve novel genes (*ATPase*, *FAB2*, *LOG2*, *MCB1*, *MOR1*, *TIV110*, *VHA*, *VLN3*, *ARM*, *Expressed*, *GET4*, *PP2A*, and *TIC110*) and six traditional genes (*EF1α*, *TUB6*, *ACT7*, *ACTIN*, *GAPDH*, and *UBC28*), were identified as candidates. After stability evaluation, two novel genes *FAB2* and *MOR1* were selected as the most optimal normalization factors during sugarcane female gametophyte development.

   *FAB2* encodes a FATTY ACID SYNTHESIS protein named stearoyl-ACP desaturase and plays an essential role in fatty acid desaturation. In *Arabidopsis thaliana*, *AtFAB2* is highly expressed in embryo and endosperm and participates in oil storage in mature seed (*Jin et al., 2017*). In *Oryza sativa*, *OsFAB2-9*, *OsFAB2-1* and *OsFAD3-1* were constitutively expressed in reproductive organs and showed relatively higher abundance (*Zhiguo et al., 2019*). For *Nicotiana benthamiana*, three *NbSACPD* genes shared high similarity to *AtFAB2*, predominantly expressed in ovules, and played essential roles in maintaining membrane composition in ovule for female fertility (*Zhang et al., 2014*). These results implied this gene and its homolog were functionally conservative during the progress of reproductive organs formation. Therefore, *FAB2*, as a stable reference gene identified in this study may also play a vital role in sugarcane female gametophyte development.

   The *MOR1* gene belongs to the ARM repeat superfamily containing CLIP-associated (CLASP) N terminal and plays an essential role in maintaining microtubule stability (*Grallert et al., 2006*). The homologous gene in yeast, *XMAP215* encodes microtubule polymerase, is associated with microtubule formation underlying chromosome segregation spatiotemporally (*Yukawa et al., 2019*). In animals including *Drosophila*, The *XMAP215* gene was also identified as the regulator of microtubule arrangement, equipping protein with unique spatial configuration and further playing numerous vital functions (*Brittle & Ohkura, 2005*). Similarly, in angiosperms, such as *Arabidopsis*

*thaliana*, *Selaginella moellendorffii*, *Volvox carteri*, *Oryza sativa*, and *Physcomitrella patens*, *MOR1* regulates the cell division and expansion *via* controlling microtubule arrays of the cell cytoskeleton. In our study, the *MOR1* gene was identified as part of stably expressed genes in sugarcane female gametophyte development, implying that the *MOR1* gene may be involved in microtubule formation followed by chromosome segregation during ovule development in sugarcane.

## CONCLUSIONS

In summary, this study is the first genome-wide reference gene identification for normalization of qRT-PCR analysis during sugarcane female gametophyte development. Based on genome and transcriptome data, all putative candidates were selected and screened by their CV values, and then the expression stabilities of eighteen putative candidates, including six traditional housekeeping genes and 12 novel genes were further evaluated by three popular algorithms. Finally, two stably expressed genes *FAB2* and *MOR1* were determined as the optimum reference genes for sugarcane female gametophyte development. Taken together, our study not only enriches the reference genes selected for the relative species but also provides the directions for studying the mechanism of the sugarcane ovule development.

## ACKNOWLEDGEMENTS

We wish to thank all members of the Qin lab for their suggestions and discussion of the results.

### Funding

This work was supported by the Science and Technology Major Project of Guangxi (Gui Ke 2018-266-Z01), the National Natural Science Foundation of China (31800262; U1605212; 31761130074; 31600249; 31700279), the China Postdoctoral Science Foundation (2018M632564), The Weng Hongwu Academic Innovation Research Fund of Peking University, and a Guangxi Distinguished Experts Fellowship. The funders had no role in study design, data collection and analysis, decision to publish, or preparation of the manuscript.

### Grant Disclosures

The following grant information was disclosed by the authors:
Science and Technology Major Project of Guangxi: Gui Ke 2018-266-Z01.
National Natural Science Foundation of China: 31800262; U1605212; 31761130074; 31600249; 31700279.
China Postdoctoral Science Foundation: 2018M632564.
Weng Hongwu Academic Innovation Research Fund of Peking University.
Guangxi Distinguished Experts Fellowship.

## Competing Interests

The authors declare that they have no competing interests.

## Author Contributions

- Maokai Yan conceived and designed the experiments, performed the experiments, prepared figures and/or tables, and approved the final draft.
- Xingyue Jin performed the experiments, prepared figures and/or tables, and approved the final draft.
- Yanhui Liu analyzed the data, prepared figures and/or tables, and approved the final draft.
- Huihuang Chen performed the experiments, prepared figures and/or tables, and approved the final draft.
- Tao Ye performed the experiments, authored or reviewed drafts of the paper, and approved the final draft.
- Zhimin Hou analyzed the data, authored or reviewed drafts of the paper, and approved the final draft.
- Zhenxia Su analyzed the data, authored or reviewed drafts of the paper, and approved the final draft.
- Yingzhi Chen analyzed the data, authored or reviewed drafts of the paper, and approved the final draft.
- Mohammad Aslam analyzed the data, authored or reviewed drafts of the paper, and approved the final draft.
- Yuan Qin conceived and designed the experiments, prepared figures and/or tables, and approved the final draft.
- Xiaoping Niu conceived and designed the experiments, prepared figures and/or tables, and approved the final draft.

## Data Availability

Raw data are available in the Supplemental Files.

The sequences are available at the Saccharum Genome Database http://sugarcane.zhangjisenlab.cn/sgd/html/index.html: MW888380 to MW888397.

## Supplemental Information

Supplemental information for this article can be found online at http://dx.doi.org/10.7717/peerj.12298#supplemental-information.

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
