# Peer review of "Identification and evaluation of the novel genes for transcript normalization during female gametophyte development in sugarcane"

_PeerJ, doi:10.7717/peerj.12298_

## Round 0.1 · original submission · Minor Revisions

The two reviewers have raised a number of specific points that should be considered, and the English should be improved before your manuscript is suitable for publication. Please submit a revised version of your manuscript together with a cover letter where you indicate how you tackled with the reviewers' comments.

Reviewer 1 ·

Basic reporting

Yan et al. manuscript identified candidate genes for normalization of qRT-PCR analysis during sugarcane female gametophyte development. Authors performed a genome wide approach by using available RNAseq data to identify suitable candidates based on coefficient of variation of average RPKM values and level of expression (highest RPKM values).
Twelve selected candidates were further analyzed, together with 6 well known, widely used normalization genes, with proper tools (geNorm, BestKeeper and NormFinder).
Selected reference genes (FAB2 and MOR1) were finally validated through the expression analysis of 2 other genes, comparing qRT-PCR and RNAseq data.

Experimental design

The experimental design is valid, and proper techniques are used.

Validity of the findings

The identification and validation of reference genes to be used for qRT-PCR normalization in specific developmental processes constitutes a very valuable tool that guarantee for accurate gene expression analysis in future studies concerning female gametophyte development in sugar cane.

Additional comments

In general, the manuscript is OK, it provides novel and useful data. However, English language and supplemental data should be improved to be accepted for publication.

The English language should be improved. Some examples where the language could be improved include lines 53, 72, 92, 100, 134, 166, 222, 251, 301– the current phrasing makes comprehension difficult.
Figures are clear and relevant, however, supplemental files need more description: in Table S1, what are the values and units? Expression levels, FKPM? In TableS2, what are the values, Cts? There is no SD shown, please, add to the table. In Table S3, add SD values to FKPM and Cts.

Minor corrections
In line 59, Saccharum spontaneum should be italicized.

Reviewer 2 ·

Basic reporting

It would be nice if all figures have the same writing font

Experimental design

No comment

Validity of the findings

No comment

Additional comments

The manuscript entitled “Identification and evaluation of the novel genes for transcript normalization during female gametophyte development in sugarcane” submitted to PeerJ describes an attempt to find suitable reference genes to be applied in studies of RT-qPCR during female gametophyte development. It is particularly important determining the suitable reference genes in the exact conditions of the experiment that one would like to determine gene expression variation of a particular gene or set of genes. The manuscript seems to be well-written, although it contains some inconsistencies that will be further indicated. If all the raised points be properly addressed and answered, an edited version of the manuscript can be published at PeerJ.
Comments
1. A lot of words in the text are hyphenated, especially at introduction. Please correct that.
2. If you are using a two-step qPCR, the correct name is ‘RT-qPCR’ and not qRT-PCR. qRT-PCR is used for one-step qPCR.
3. Please, define AC and MMC.
4. Avoid ‘strong’ words when describing your results or your scientific problem. For example, on line 58 (desperately) and on line 64 (outstanding), etc.
5. On line 110 put Saccharum spontaneum in italics.
6. It would be nice a figure showing the morphology of the five different stages of female gametophyte development.
7. On line 225 the reference is duplicated.
8. On Figure 4 it would be better if the relative expression were transformed to log, to see the expression values between 0 and 1.
9. This is a manuscript about finding suitable references genes in a given condition. It is nice that the authors provided as supplemental material all the information about primer efficiency. However, it is important to show the melting curves for each primer. This information would help other researchers to replicate the experiments using the same set of primers.
10. Also, at table S2 what are those values? Ct values?

---

## Round 0.2 · Minor Revisions

The previous Academic Editor is no longer available and so I have taken over handling your submission.

After reading the manuscript several grammar issues were noted, yet appeared readily repairable. I have listed the suggested changes below. I did notice that you made note of some of the functionality of the newer defined candidate reference genes in relation to their function. It may be worthwhile assigning gene ontology terms to these if they may share similar functional annotations; this would be optional. Otherwise, the content appeared rather straightforward and useful. I will suggest only minor modifications before moving the manuscript forward. Congratulations on your efforts.

Example of annotation:
LINE NO.: / PREVIOUS FORM / SUGGESTED FORM / [ADDITIONAL NOTES, NONE [.]]
LINE 45.: / harvested / with harvested yield at / [.]
LINE 57.: / is an vital / is a vital endeavor / [.]
LINE 59.: / signaling / signal / [.]
LINE 60.: / transduction that underlie during the female / transduction networks during female / [.]
LINE 67.: / and so on / among other factors / [.]
LINE 71.: / showed / demonstrate / [.]
LINE 72.: / depends on the experimental condition of tissue / depend on the experimental conditions of the tissue / [.]
LINE 77.: / researches / research studies / [.]
LINE 84.: / such as apple / such as with apple / [.]
LINE 84.: / sexual organ / sexual organs / [.]
LINE 84.: / peer peel / pear peel / [.]
LINE 88.: / peer peal / pear peel / [.]
LINE 108: / Megespore mother cell / megaspore mother cell / [.]
LINE 113: / by the gel electrophoresis / by gel electrophoresis / [.]
LINE 114: / was referred to the / was referred to by the / [.]
LINE 115: / Each sample has three / Each sample had three / [.]
LINE 123: / primer designing / primer design / [.]
LINE 136: / experiment as follows / experiment was as follows / [.]
LINE 141: / gene is stable / gene was stable / [.]
LINE 142: / software, / software packages, / [.]
LINE 144: / genes can be calculated / genes were calculated / [.]
LINE 153: / across 5 different / across five different / [.]
LINE 153: / sugarcane ovule AC / sugarcane ovule at AC / [.]
LINE 165: / housekeeping gene’s expression / housekeeping gene expression levels/ [.]
LINE 175: / genes have a relatively / genes had a relatively / [.]
LINE 176: / experiment have high / experiment had high / [.]
LINE 178: / each gene were calculated / each gene was calculated / [.]
LINE 179: / and R2 / and the R2 / [.]
LINE 185: / The lower Ct value means the higher expression / The lower the Ct value means the higher the expression / [.]
LINE 191: / , showed / , which showed / [.]
LINE 208: / value are negatively / value were negatively / [.]
LINE 210: / with the lowest (CV / with the lowest value (CV / [.]
LINE 230: / Reference Genes Validation / Reference Gene Validation / [.]
LINE 239: / and reduced at the / and was reduced at the / [.]
LINE 243: / that of the most genes / that of most genes / [.]
LINE 248: / 2020). / 2020) / [remove period.]
LINE 248: / heterogeneity reduced / heterogeneity being reduced / [.]
LINE 250: / of sugarcane female / of the sugarcane female / [.]
LINE 258: / reference genes selection / reference gene selections / [.]
LINE 262: / suitable genes under / suitable gene under / [.]
LINE 267: / Sorghum mosaic virus / Sorghum Mosaic Virus / [.]
LINE 290: / XMAP215 gene / the XMAP215 gene / [.]
LINE 295: / of cell cytoskeleton. / of the cell cytoskeleton. / [.]
LINE 296: / as stably expressed genes / as part of stably expressed genes / [.]
LINE 296: / that MOR1 / that the MOR1 / [.]
LINE 297: / gene may involve in / gene may be involved in / [.]

---

## Round 0.3 · accepted · Accept

Thank you for addressing the suggested edits. I do think that additional ontology annotations would have added impact value to the manuscript; however, it was suggested as optional. I will agree that the manuscript is in shape to move forward. Congratulations on your efforts.